# OsSHMT4 Is Required for Synthesis of Rice Storage Protein and Storage Organelle Formation in Endosperm Cells

**DOI:** 10.3390/plants13010081

**Published:** 2023-12-26

**Authors:** Mengyuan Yan, Ziyue Zhou, Juling Feng, Xiuhao Bao, Zhengrong Jiang, Zhiwei Dong, Meijie Chai, Ming Tan, Libei Li, Yaoliang Cao, Zhanbo Ke, Jingchen Wu, Zhen Feng, Tian Pan

**Affiliations:** 1The Key Laboratory for Quality Improvement of Agricultural Products of Zhejiang Province, College of Advanced Agricultural Sciences, Zhejiang A&F University, Hangzhou 311300, China; yanmengyuan@zafu.edu.cn (M.Y.); 2022101012024@stu.zafu.edu.cn (Z.Z.); 2021101011013@stu.zafu.edu.cn (Z.D.); 2022101012002@stu.zafu.edu.cn (M.C.); 2023101011012@stu.zafu.edu.cn (M.T.); libeili@zafu.edu.cn (L.L.); 2022201011001@stu.zafu.edu.cn (Y.C.); 202120020224@stu.zafu.edu.cn (Z.K.); gvpiaoye@stu.zafu.edu.cn (J.W.); 2College of Agronomy, Northwest A&F University, Yangling 712100, China; 2022060037@nwafu.edu.cn; 3Institute of Crop Sciences, Ningbo Academy of Agricultural Sciences, Ningbo 315000, China; baoxiuhao123@163.com; 4College of Agronomy, Nanjing Agricultural University, Nanjing 210095, China; jzr@njau.edu.cn

**Keywords:** seed storage protein, *flo20-1*, PBII, PBI, transcription levels

## Abstract

Storage proteins are essential for seed germination and seedling growth, as they provide an indispensable nitrogen source and energy. Our previous report highlighted the defective endosperm development in the *serine hydroxymethyltransferase 4* (*OsSHMT4*) gene mutant, *floury endosperm20*-*1* (*flo20-1*). However, the alterations in storage protein content and distribution within the *flo20-1* endosperm remained unclear. Here, the immunocytochemistry analyses revealed a deficiency in storage protein accumulation in *flo20-1*. Electron microscopic observation uncovered abnormal morphological structures in protein bodies (PBI and PBII) in *flo20-1*. Immunofluorescence labeling demonstrated that aberrant prolamin composition could lead to the subsequent formation and deposition of atypical structures in protein body I (PBI), and decreased levels of glutelins and globulin resulted in protein body II (PBII) malformation. Further RNA-seq data combined with qRT-PCR results indicated that altered transcription levels of storage protein structural genes were responsible for the abnormal synthesis and accumulation of storage protein, which further led to non-concentric ring structural PBIs and amorphous PBIIs. Collectively, our findings further underscored that *OsSHMT4* is required for the synthesis and accumulation of storage proteins and storage organelle formation in endosperm cells.

## 1. Introduction

Seed proteins serve as a source of the nitrogen and sulfur required for developing seedlings. More than 90% of a rice grain consists of starch and proteins, and the grain protein content (GPC) is a crucial indicator of the grain’s nutritional quality [1]. Although rice is generally acknowledged to have the lowest GPC, net protein utilization from rice is the highest among cereal grains. There will likely be a significant rise in the demand for premium cereals as a source of protein in the near future. Based on their functional roles, proteins can be classified into two groups: structural proteins, also known as housekeeping proteins, which support the regular metabolism of seed cells, and seed storage proteins (SSPs), which serve as organic nitrogen and amino acid reserves for embryonic and seedling development. The second most abundant form of storage material after endosperm starch in rice seeds is storage protein, which typically makes up 8% to 10% of the dry weight of brown rice. Compared to other cereal crops, rice has a greater ratio of digestible protein to accessible protein, and its storage protein is readily absorbed and used by the body. Additionally, SSPs are abundantly accumulated in the peripheral aleurone/subaleurone and the inner starchy endosperm of rice. Rice seed storage proteins can be categorized into four groups based on their varying solubility: water-soluble albumin, salt-soluble globulin, alcohol-soluble prolamin, and alkali-soluble glutelin [2]. Of these, rice glutelin makes up the greatest amount (more than 60%) and is followed by prolamins (20%) in the overall amount of storage protein, which is different from the alcohol-soluble seeds of other cereal crops.

Glutelins and prolamins are both produced on the rough endoplasmic reticulum (ER), but they concentrate in distinct subcellular compartments depending on the highly dynamic and functionally specialized endomembrane system [3,4,5,6]. Glutelins are initially synthesized on the rough ER and subsequently introduced into the vesicular-mediated transport system, rendering their accumulation in the protein storage vacuoles (PSVs) as electron-dense and irregularly shaped granules known as protein body II (PBII) with a diameter of 2–4 μm through the Golgi apparatus and post-Golgi trafficking pathway [3,7,8,9,10,11]. Based on the similarity of their amino acid sequences, glutelins may be further classified into four families (GluA, GluB, GluC, and GluD) [12]. Glutelins are initially produced as a 57 kDa precursor and cleaved into a 37–39 kDa acidic subunit and a 22–23 kDa basic subunit in the cytoplasm. In addition, glutelins are found within the central region of PBIIs, where they form crystalline structures with lattice patterns, whereas α-globulin is predominantly situated in the outer matrix that envelops the glutelins [13].

The prolamins, with the help of molecular chaperone BiP, aggregate to form spherical protein body I (PBI) within the ER lumen, which is 1–3 μm in diameter and has a concentric ring structure [3,14,15,16]. SDS-PAGE analysis of isolated PBIs showed that they contain 10 kDa (Pro10), 13 kDa (Pro13), and 16 kDa (Pro16) prolamins [17]. There are two distinct groups of 13 kDa prolamins: 13b prolamin was extracted without the use of a reducing reagent, while 13a prolamin was extracted using reducing conditions akin to those applicable to the 10 and 16 kDa prolamins [17]. Among them, Pro10, Pro13a, and Pro16 are rich in sulfur-containing amino acids (CysR), while Pro13b is poor in sulfur-containing amino acids (CysP) [18]. PBIs consist of 60% CysR prolamins and 40% CysP prolamins [17,19]. These prolamin genes exhibit highly spatially and temporally specific expression patterns in the developing endosperm, resulting in heterogeneous protein distribution and a unique PBI structure in the rice endosperm. PBI is composed of an exterior layer rich in Pro13b composition, a middle layer rich in Pro13a and Pro16, an interior layer rich in Pro13b, and a Pro10 core [20]. 

The nutritional value of rice glutelin is superior to that of other rice SSPs due to its higher lysine content and greater digestibility in humans. However, reducing the content of digestible glutelin is beneficial for those with renal illness, who require low protein intake [21]. Thus, the balance of glutelin and prolamin in the rice endosperm is noteworthy. In a previous study, we reported seven allelic floury endosperm mutants, *flo20-1* to *flo20-7*, which were caused by a single-nucleotide substitution in *serine hydroxymethyltransferase 4* (*OsSHMT4)* [22]. We chose *flo20-1* for in-depth studies and found that the *flo20-1* endosperm was filled with loosely arranged and abnormal starch granules as well as amorphous PBIIs [22]. However, the specific protein bodies’ morphologies in the *flo20-1* endosperm remain unclear. Therefore, in this study, we demonstrated the morphological structures of PBIs and PBIIs in *flo20-1* endosperm through cytological experiments to uncover the explicit effects of OsSHMT4 on seed storage proteins, and we performed an RNA-seq analysis to elucidate underlying causes. 

## 2. Results

### 2.1. Storage Protein Composition Changed in the flo20-1 Mutant

To better compare the SSP composition changes in *flo20-1*, we extracted the total storage proteins from mature wild-type and *flo20-1* seeds and conducted SDS-PAGE and immunoblot analyses. In the wild type, the major storage proteins consist of the 40 kDa acidic and 20 kDa basic glutelin subunits, 26 kDa globulin polypeptides, 10 to 16 kDa prolamin polypeptides, and small amounts of the 57 kDa glutelin precursor (Figure 1A, left lane). In contrast, the *flo20-1* mature seeds exhibited slightly lower amounts of the 40 kDa acidic and 20 kDa basic subunits of the mature glutelin and a significant reduction of 26 kDa α-globulin in comparison to the wild-type seeds (Figure 1A, right lane). In addition, 16 kDa and 10 kDa prolamins were both notably declined, while the 13 kDa prolamin was markedly increased in the *flo20-1* seed (Figure 1A, right lane). To confirm the SSP composition changes observed in *flo20-1*, we performed an immunoblot analysis using specific polyclonal antibodies against the 40 kDa acidic glutelin subunit, globulin, and prolamins. Immunoblots using isoform-specific antibodies confirmed a decrease in the accumulation of acidic subunits across all glutelin subfamilies (GluA, GluB, GluC, and GluD) along with a significant decrease in the accumulation level of α-globulin (Figure 1B). It is interesting to note that the *flo20-1* seed had an evident decrease in Pro16, Pro13a, and Pro10 but a large increase in Pro13b in comparison to the wild type (Figure 1C). These results suggest that *flo20-1* is defective in storage protein accumulation.

To explore whether *flo20-1* is defective in storage protein sorting at the ER level or in post-Golgi trafficking, we assessed the expression levels of two molecular chaperones, BINDING PROTEIN1 (BiP1) and PDI1-1, which play an essential role in the segregation and aggregation of proglutelin and prolamin polypeptides within the ER lumen [7,15,23]. The immunoblot analysis revealed comparable levels of BiP1 and PDI1-1 in both the *flo20-1* mutant and the wild type (Figure 1D), indicating that *flo20-1* does not enact quality control within the ER. These observations further suggested that the phenotypes observed in the *flo20-1* mutant were likely attributable to a defect in the regulatory factor OsSHMT4 rather than the structural genes of SSPs.

### 2.2. Amorphous Structure of PBII in the flo20-1 Mutant

To directly uncover the morphological structure of SSPs, we prepared ultrathin sections of developing endosperms at 9, 15, and 21 DAF and conducted transmission electron microscopy (TEM) observation to compare subcellular structures between wild-type and *flo20-1* mutants. PBII is a vacuolar organelle that deposits glutelins and globulins. As shown in Figure 2A, the vacuolar membrane was easily detected around PBII at the early stage of endosperm development in the wild type. At 9 DAF, the morphology of PBIIs was largely comparable in the wild-type and *flo20-1* mutants (Figure 2A,B). Afterwards, PBIIs were filled with storage proteins and appeared as structures with irregular shapes and dense staining at 15 DAF (Figure 2C). At 21 DAF, PBIIs accumulated in the gaps between starch granules (Figure 2E). However, the structure of deformed PBII in *flo20-1* seemed to be different from deconstructed, wild-type PBII during the late stage of grain filling. The patterns of PBIIs in *flo20-1* were more misshapen than those of the wild type after 15 DAF (Figure 2D,F). 

Previous research has suggested that PBII formation and/or development may be under feedback regulation according to PBII protein amounts [24]. To further verify the subcellular distribution of storage proteins, we prepared and subjected the semi-thin sections of wild-type and *flo20-1* developing endosperm to double immunofluorescence labeling with specific antibodies against globulins and glutelins. In the wild type, α-globulins were predominantly localized to the periphery of PBIIs, surrounding the glutelins (Figure 3A), while the immunofluorescence signal of α-globulin was almost invisible in *flo20-1* (Figure 3B). The form of PBII was consistent with the TEM observations (Figure 2), which is similar to the previously reported 26 kDa globulin-deficient mutant [25,26]. It is possible that the localization of globulins in the matrix of PBII is essential to keep the normal morphology of PBII. These findings suggested that the *OsSHMT4* mutation led to a large decrease in globulin concentration, which impacted PBII morphogenesis and resulted in PBII malformation.

### 2.3. Abnormal Morphological Structure of PBI in the flo20-1 Mutant

Rice prolamins are synthesized on the rough ER and form PBIs in endosperm cells. In order to obtain more information about the morphological characteristics of PBI in *flo20-1*, thick sections of freshly harvested seeds (18 DAF) stained with rhodamine were adopted for subcellular observation, as the content of prolamins in the *flo20-1* mutant significantly changed. As shown in Figure 4A, round-shaped PBI can be clearly observed in both wild-type and *flo20-1* endosperms. But, the size of rhodamine-labeled PBI in the *flo20-1* mutant was smaller than in the wild type. We further measured the diameters of rhodamine-labeled PBI and the number of rhodamine-labeled PBI per 100 μm^2^. The statistical results showed that the diameter of PBI in *flo20-1* was nearly half as small as in the wild type, and the number of PBIs also decreased markedly (Figure 4B,C). 

To investigate whether there were any differences in the PBI maturation process between the wild-type and *flo20-1* endosperms’ sub-aleurone cells, a TEM of the developing rice seeds was performed. In the wild type, PBIs of 9 DAF endosperm were spherical structures consisting of intensely stained material surrounded by a peripheral ring of lightly stained material (Figure 5A). At 15 DAF, the PBIs contained a concentric structure of alternating light and dark layers (Figure 5C). At 21 DAF, the PBIs had concentric rings of varying electron density and were larger than those in the early stages (Figure 5E). The representative concentric ring structures of PBIs were observed in the wild-type endosperm during different developmental stages, whereas there were no obvious changes in *flo20-1* (Figure 5B,D,F). The PBI displayed uniform electron density in the mutant, rather than the concentric structure of alternating light and dark layers in the wild type, which may be caused by the defection in prolamin composition. 

To trace the distribution of prolamins, double immunofluorescence labeling was carried out with targeted antibodies directed against the glutelins (anti-acid subunits) and prolamins (anti-Pro10 and anti-Pro13b) in subaleurone cells of developing endosperm at 9 DAF (Appendix A). Notably, we could not find the immunofluorescence signal of Pro10 in the *flo20-1* mutant (Appendix A), while the abnormal PBIIs were consistent with previous observations (Figure 2 and Figure 3). Subsequently, different specific antibodies against Pro10, 13a, and 13b were applied to further clarify the positional relationship of prolamins in view of the distinct immunofluorescence signal of Pro10 and Pro13b in the wild-type and *flo20-1* mutants. Using a combination of anti-Pro10 and anti-Pro13b prolamin antibodies revealed that Pro10 was mainly detected in the core of PBIs, whereas Pro13b was detected in a ring surrounding the 10 kDa prolamin-rich core of PBIs in the wild-type endosperm at 21 DAF (Figure 6A), which was consistent with former reports [20]. To the contrary, the *flo20-1* mutant lacked the immunofluorescence signal of Pro10 and did not exhibit an obvious PBI ‘doughnut structure’. The combination of the anti-Pro13b and anti-Pro13a antibodies showed that Pro13a was mostly located within a layer between the Pro13b-rich layers in the wild type (Figure 6B). What is noteworthy is that the concentric ring structure was not observed in the *flo20-1* mutant. It seemed that the distribution of Pro13b was disordered in *flo20-1*. Together, these observations suggested that aberrant prolamin composition in *flo20-1* resulted in the subsequent formation and deposition of abnormal structures of PBI. 

### 2.4. Transcriptome Sequencing Analysis of the flo20-1 Endosperm

To further elucidate the molecular mechanisms underlying the phenotypic differences between *flo20-1* and the wild type, we performed high-throughput sequencing and data analysis of RNA extracted from 9 DAF developing endosperm of the WT and *flo20-1*. Six cDNA libraries with three biological replicates were sequenced. After filtering out any low-quality sequences and adaptors, a clean data set totaling 40.44 Gb was obtained, for which the average clean reads per sample were 4.49 million, together with no less than 92.49% of the Q30 ratio and over 49.66% of the GC content (Appendix A). Through alignment analysis, 91.86–96.12% of the clean data were mapped on the genome of *Oryza sativa*, which implied a reliable quality of the RNA-seq (Appendix A). 

In total, 18,263 and 18,442 expressed genes were identified in the wild type and *flo20-1*, respectively, with 17,412 genes expressed in both materials (Figure 7A). A volcano readily showed that 1402 genes (log_2_ fold changes greater than 1, padj < 0.05) in *flo20-1* were differentially expressed compared to that of the wild type, of which 727 genes were downregulated while 675 genes were upregulated (Figure 7B). To understand the function of the differentially expressed genes (DEGs), their KEGG enrichment analysis was performed. The DEGs were enriched to several essential biological pathways mainly related to DNA replication and repair, amino acid metabolism and metabolite synthesis, including DNA replication, cutin, suberine, and wax biosynthesis, nitrogen metabolism, galactose metabolism, glycerolipid metabolism, protein processing in endoplasmic reticulum, phenylpropanoid biosynthesis, cysteine and methionine metabolism, homologous recombination, pentose and glucuronate interconversions, mismatch repair, starch and sucrose metabolism, and glutathione metabolism (Figure 7C). More strikingly, DNA replication was the most enriched pathway, thus implicating its vital role in the regulation of endosperm development (Figure 7C). 

### 2.5. Analysis of DEGs Related to the SSP Synthesis

Owing to the fact that the content of storage protein components changed in *flo20-1* according to SDS-PAGE and immunoblot analyses, we searched and compared the expression levels of storage protein structural genes from the transcriptome data (Appendix A). As shown in Figure 8A, compared to the wild type, the expression levels of several Pro13b structural genes (Pro13b.5, Pro13b.8, Pro13b.10, Pro13b.14, Pro13b.15, Pro13b.16, Pro13b.17, Pro13b.18, Pro13b.19, Pro13b.20, Pro13b.21, and Pro13b.22) were prominently elevated in *flo20-1*. Adversely, the expression levels of glutelin genes (GluA1, GluA2, and GluC1), a globulin gene (α-globulin), Pro10 genes (Pro10.2 and Pro10.3), and Pro13a genes (Pro13a.1, Pro13a.3, Pro13a.5, and Pro13a.6) were evidently decreased. Eighteen DEGs in the RNA-seq data were selected for validation using qRT-PCR, and the results showed that the relative expression levels of DEGs according to qRT-PCR corresponded well with the RNA-seq data (Figure 8B). These analyses indicated that the alteration of transcription levels of storage protein structural genes was responsible for the abnormal synthesis and accumulation of storage protein, which further led to atypical PBIs and amorphous PBIIs. 

## 3. Discussion

Rice endosperm development is regulated by multiple subcellular organelles and genes, including those involved in starch synthesis and storage protein biosynthesis regulation. In a previous study, we had identified a floury *flo20-1* mutant with defective endosperm development, along with abnormal accumulation of starch and storage proteins [22]. In this study, we further discovered that the *flo20-1* mutant showed different content of the storage protein component compared with the wild type, a slight decrease in acidic subunits of glutelin, a remarkable increase in Pro13b, and a distinct reduction in Pro10, Pro13a, Pro16, and 26 kDa α-globulin. It is likely that there existed complementary mechanisms controlling the availability of free amino acids, thus resulting in decreases in several SSP components accompanied by increases in the amount of another.

Studies on genes involved in protein-processing, protein-sorting, or protein-trafficking mutants have revealed that their mutations affected PB structures. The mutation of PDI1 caused large amounts of PBI-like structures containing cross-linked glutelins and prolamins in developing endosperm [7]. The mutation of vacuolar processing enzyme (VPE1) developed round-shaped PBIIs [27,28]. The *gpa4* mutant possessed immature PBIs and novel structures with a glutelin core and a prolamin periphery in the developing endosperm [10]. The other mutants defective in post-Golgi trafficking, *gpa1*-*gpa3* and *gpa5-gpa8*, all developed large amounts of protein granules and extracellular paramural bodies filled with glutelins and globulins [8,9,11,29,30,31,32]. Our cytological and immunocytochemical studies manifested that there were novel PBIs without concentric ring structures and disformed PBIIs in *flo20-1*. These structures are quite different to those observed in previously reported mutants. Notably, two types of abnormal PB structures in *flo20-1* were very similar to the morphology observed in the mutants with altered SSP compositions [24,25]. 

Given that α-globulin plays a key part in deposition and storage together with glutelin within the peripheral region of the PBIIs, its spatial arrangement in PBIIs is essential. In micrographs, PBIIs in GluB·Glb-less transgenic plants looked cracked, indicating that the PBII structure was not sufficiently maintained by filling it with glutelins alone [24]. In the globulin-less mutant, glutelins were improperly packed into PBIIs, which resulted in the disordered spatial structure of PBIIs, thus indicating the vital role of α-globulin in frame constructing during PBII formation [25]. Beyond that, the PBIs in *globulin-RNAi* seeds presented an uneven or dull outline, and a deficiency in globulin may affect the quantity and stability of other storage proteins during seed development [26]. These observations indicated that a lower content of globulin would hinder the intracellular structures of the developing endosperm cells. It is worth noting that the phenotype of PBIIs observed in *flo20-1* is almost the same as in the α-globulin knockdown mutant. In our studies, the concentration of α-globulin dropped dramatically, and its immunofluorescence signal disappeared in *flo20-1*. Thus, we concluded that the collapse of PBIIs in *flo20-1* may be under feedback regulation by declined glutelins and globulin amounts.

A typical sphere structure of mature PBIs is that Pro10 is mainly localized in the core of the PBIs, Pro13b is localized in the inner layer surrounding the core and the outermost layer, and Pro13a and 16 are localized in the middle layer [18,20]. Changes in prolamin components could cause changes in the structure or morphology of PBIs. Electron microscopic observations demonstrated that *esp1* and *esp4*, both with a decrease in Pro13b, had quite similar PBI structures to the wild type [33]. In contrast, no typical PBI structure was seen in the *esp3* mutant with reduced Pro10 and Pro13a levels [33]. In Pro10-less, Pro13a-less, and Pro16-less transgenic plants, rhodamine-labeled PBIs could barely be detected, but the degree of these alterations was much lower in Pro16-less plants than for Pro13a-less and Pro10-less plants, reflecting the notion that Pro16 is a minor component [24]. These findings showed that differences in PBI composition arising from polypeptide differences were further embodied in PBI structures, and Pro10 and Pro13a played a vital role in the association of other prolamin polypeptides and the typical structure formation of PBIs. It is notable that the concentric ring structure of PBI was invisible in the *flo20-1* mutant, and instead was replaced by electron-lucent and uniformly electron-dense prolamin PBs. Our results showed that there was an evident reduction in Pro16, Pro13a, and Pro10 but a large increase in Pro13b in *flo20-1*. Therefore, we deduced that the absence of a concentric ring structure in PBIs in *flo20-1* was caused by the smaller amount of Pro13a and Pro10 polypeptides. 

To assess whether variations in SSP accumulation are reflected in changed mRNA levels, we conducted RNA-seq and qRT-PCR analyses using RNA extracted from developing seeds at 9 DAF. As expected, these mRNA expression levels of several SSP synthesis genes basically paralleled the SSP accumulation levels, indicating that the compensatory mechanism for decreased SSPs, achieved by augmenting other SSPs, is primarily regulated at the transcriptional level. In addition to these SSP structural genes, some DEGs were involved in starch and sucrose metabolism according to the KEGG analyses, which explained the abnormal accumulation of starch content in *flo20-1*. The serine hydroxymethyltransferase catalyzes the reversible interconversion of serine and glycine and is related to sulfur metabolism [34]. We had reported that OsSHMT4 works cooperatively with proteins like OsSHMT3 to execute SHMT activity [22]. Given that the content of CysR SSPs, including glutelins, globulin, Pro10, Pro13a, and Pro16, had declined while CysP Pro13b was promoted in *flo20-1*, it is not difficult to understand that DEGs were involved in cysteine and methionine metabolism. In mammalian cells, SHMT1 and SHMT2α serve as essential scaffold proteins, which are required for translocating to the nucleus for DNA replication and repair [35,36]. In this study, DEGs participating in DNA replication were significantly enriched. However, whether and how the nuclear-localized OsSHMT4 in rice endosperm cells contributes to the biological process of DNA replication remains unclear, which deserves to be investigated in future studies.

## 4. Conclusions

Endosperm, serving as the primary reservoir of starch and proteins in cereal crops, stands as a pivotal source of calories and essential nutrition. Given its agricultural significance and genetic diversity, rice endosperm has served as a fundamental model for unraveling the intricacies of endosperm development. Storage proteins play a critical role in seed germination and seedling growth by providing indispensable nitrogen and energy. Our prior investigation shed light on the impaired development of endosperm in the *OsSHMT4* gene mutant known as *floury endosperm20-1* (*flo20-1*), which is characterized by anomalous accumulations of starch and storage proteins. Here, through comprehensive cytological, immunocytochemical, and biochemical examinations, our studies revealed that the collapse of Protein Body II (PBII) structures in *flo20-1* might be subject to feedback regulation due to decreased levels of glutelins and globulins. Additionally, the absence of concentric ring structures in Protein Body I (PBI) in *flo20-1* was attributed to reduced amounts of Pro13a and Pro10 polypeptides. Omics analyses further elucidated that OsSHMT4 influenced the transcription levels of structural genes related to storage proteins, consequently causing aberrant synthesis and accumulation of storage proteins. This disruption in turn led to atypical PBIs and amorphous PBIIs. The composition of storage proteins significantly influences the nutritional and functional qualities of cereal crops. Our observations indicated that the loss of OsSHMT4 function disrupted the composition of storage proteins, suggesting the potential significance of SHMT4 in breeding rice varieties tailored to specific nutritional requirements. Collectively, our study established a functional association between SHMT proteins and the synthesis of storage proteins as well as the formation of storage organelles within endosperm cells.

## 5. Materials and Methods

### 5.1. Plant Materials

Experiments were performed using the *flo20-1* mutant, which is derived from an ethylmethanesulfonate-mutagenized pool of the *japonica* rice variety Kitaake [22]. Developing seeds and mature seeds were taken from plants growing in paddy fields during the normal growing season in Beijing.

### 5.2. SSP Extraction

Total seed protein extraction and immunoblot assays were performed as described previously [29]. Briefly, developing or mature dry seed proteins were extracted using 0.125 M of Tris-HCl, 4% (*w*/*v*) SDS, 4 M of urea, and 5% (*v*/*v*) 2-mercaptoethanol, pH 6.8. Sodium dodecyl sulfate–polyacrylamide gel electrophoresis (SDS-PAGE) was performed to separate the protein bands. After electrophoresis, the SDS-PAGE gels were stained with 0.1% (*w*/*v*) Coomassie Brilliant Blue R-250, 25% (*v*/*v*) isopropanol, and 10% (*v*/*v*) acetic acid and subsequently de-stained with a solution of 5% (*v*/*v*) ethanol and 10% (*v*/*v*) acetic acid. SDS-PAGE assays were repeated at least three times and pictured using the ChemiDoc System (BIO-RAD, Hercules, CA, USA).

### 5.3. Immunoblot Analyses

An immunoblot assay was performed according to a previous study [9,10]. SDS-PAGE-separated proteins were electro-transferred to a nitro-cellulose membrane, which was subsequently incubated with a 1:1000–3000 dilution of primary antibodies raised against each protein and a 1:2000 dilution of secondary antibody, goat IgG against mouse or rabbit IgG conjugated with HRP in 10 mM of PBS, 0.05% (*v*/*v*) TWEEN20 (VWR Chemicals, Radnor, PA, USA), and 5% (*w*/*v*) skim milk for 1 h each. The protein gel blot bands were visualized using ECL Plus reagent (Odyssey-Fc, LI-COR, Lincoln, NE, USA) and determined using the ChemiDoc System (BIO-RAD, Hercules, CA, USA).

### 5.4. Antibodies

Anti-glutelin acid subunits, anti-α-globulin, anti-GluA, anti-GluB, anti-GluC, anti-GluD, anti-Pro16, anti-Pro13a, and anti-Pro10 antibodies were used as described previously [9,10,11]. A peptide of Pro13b (PRYYGAPSTITTLGGVL) [20] was synthesized and then injected into rabbits and mice to generate polyclonal and monoclonal antibodies at the ABclonal Biotechnology Company, respectively (https://www.abclonal.com.cn/, 25 October 2017). All of the antibodies were used in 1:1000 dilutions in immunoblot analyses. Anti-EF-1α (Agrisera, AS10934, Uppsala, Sweden, dilution 1:3000) antibodies were purchased commercially.

### 5.5. Microscopy Observation

The rhodamine staining analyses of thick sections, immunofluorescence analyses of semithin sections, and transmission electron microscope analyses of ultrathin sections were performed as described previously [8,9,11,29].

For rhodamine staining analyses, fresh grains developing to the middle and late stages were prepared for thick sections (60 μm in thickness) through oscillating microtome (LEICA VT1200S, Wetzlar, Germany) and then fixed in buffer containing 50 mM of PIPES, 10 mM of EGTA, 10 mM of MgSO_4_, 1% (*v*/*v*) DMSO, 0.1% (*v*/*v*) Triton X-100, and 4% (*w*/*v*) paraformaldehyde for 30 min [9]. The thick sections were dyed for 10 min with 1 μM of rhodamine (Sigma-Aldrich 83689, St. Louis, MO, USA) at 37 °C and then rinsed three times with Tris-buffered saline-Tween (TBST (10 mM of Tris-HCl, 150 mM of NaCl, and 0.05% (*v*/*v*) Tween 20, pH 7.4), followed by confocal imaging using an LSM980 laser scanning confocal microscope (Carl Zeiss, Oberkochen, Germany). 

For double immunofluorescence analysis, the semithin sections (0.4 μm in thickness) of developing grains were obtained according to Ren et al. [9]. The sections used to analyze the internal structures of PBIs were prepared from 21 days after flower (DAF) grains, and all other sections were from 9 DAF developing grains. Sections were blocked in TBST containing 3% (*w*/*v*) bovine serum albumin (BSA) for 1 h and then incubated with the primary antibodies (anti-glutelin acidic subunits with 1:100 dilution, anti-prolamin with 1:50 dilution, and anti-α-globulin with 1:100 dilution) for 1 h in TBST containing 1% (*w*/*v*) BSA. After washing three times with TBST, the sections underwent 1 h of incubation with Alexa Fluor 488 (green) and/or Alexa Fluor 555 (red) conjugated secondary antibodies (Invitrogen, Carlsbad, CA, USA) at a 1:200 dilution. Subsequently, the sections were subjected to three additional TBST washes before proceeding to confocal imaging. The Alexa Fluor 488 (green) signals, emitted in the wavelength range of 495–530 nm, were captured using a white light laser with an excitation wavelength of 488 nm. Simultaneously, Alexa Fluor 555 (red) signals falling within the emission range of 561–596 nm were recorded using a white light laser with an excitation wavelength of 555 nm.

For the transmission electron microscope analysis, ultrathin sections (50–70 nm in thickness) of 9 DAF, 15 DAF, and 21 DAF endosperm from WT and *flo20-1* were prepared as described previously [8,9,10,11,29,32]. After post-staining with aqueous uranyl acetate/lead citrate, the samples were examined using a Hitachi H7700 transmission electron microscope (Tokyo, Japan). 

### 5.6. RNA Extraction and RNA-seq Analysis

The total RNA was extracted from 9 DAF endosperm of WT and *flo20-1* with three biological replicates using an RNA Prep Pure kit (TIANGEN Biotech, Beijing, China) following the manufacture’s recommendations. RNA samples were sent to Novogene Bioinformatics Institute (Beijing, China) for library construction, high-throughput sequencing, and data analysis using an Illumina HiSeq 2500 platform. Fragments per kilobase million reads (FPKM) were used to quantify levels of gene expression. The genes with |log_2_FoldChange| ≥ 1 and padj ≤ 0.05 were considered differentially expressed genes (DEGs). Functional classification and pathway analysis of the obtained DEGs were performed through Kyoto Encyclopedia of Genes and Genomes (KEGG) analysis. 

### 5.7. Quantitative Real-Time PCR (qRT-PCR) Validation of DEGs

Eighteen DEGs associated with SSP synthesis were validated through qRT-PCR analysis. For the qRT-PCR analysis, the first-strand cDNA was synthesized from 2 μg of total RNA based on a PrimeScript Reverse Transcriptase Kit (TaKaRa, Kyoto, Japan). qRT-PCR was performed in an ABI7500 Real-time PCR system (Thermo Fisher Scientific, Waltham, MA, USA) using the SYBR Premix Ex Taq (TaKaRa, Kyoto, Japan) with rice Ubiquitin as the internal control. The relative quantification in gene expression levels was calculated based on three biological replicates by referring to the 2^−∆∆Ct^ method [37]. Specific primers used for qRT-PCR are listed in Appendix A.

## Figures and Tables

**Figure 1 plants-13-00081-f001:**
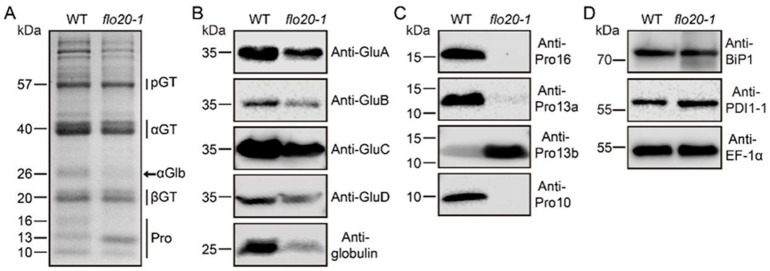
Storage protein phenotypes of the *flo20-1* mutant. (**A**) SDS-PAGE storage protein profiles of dry seeds were compared with the wild-type (WT) and *flo20-1* mutant using an SDS gel stained with Coomassie Brilliant Blue (CBB). pGT, unprocessed glutelin precursors; αGT, mature glutelin acidic subunits; αGlb, α-globulin; βGT, mature glutelin basic subunits; Pro, prolamins. (**B**) Immunoblot analysis of the glutelin subfamily proteins (GluA, GluB, GluC, and GluD) and α-globulin from dry seeds of WT and *flo20-1* mutant using subfamily-specific antibodies. (**C**) Immunoblot analysis of prolamin subfamily proteins from dry seeds using anti-prolamin subfamily-specific (Pro10, Pro13a, Pro13b, and Pro16) antibodies. (**D**) Immunoblot analysis was performed on dry seeds using anti-molecular chaperone antibodies (BiP1 and PDI1-1). EF-1α served as the loading control in (**B**–**D**).

**Figure 2 plants-13-00081-f002:**
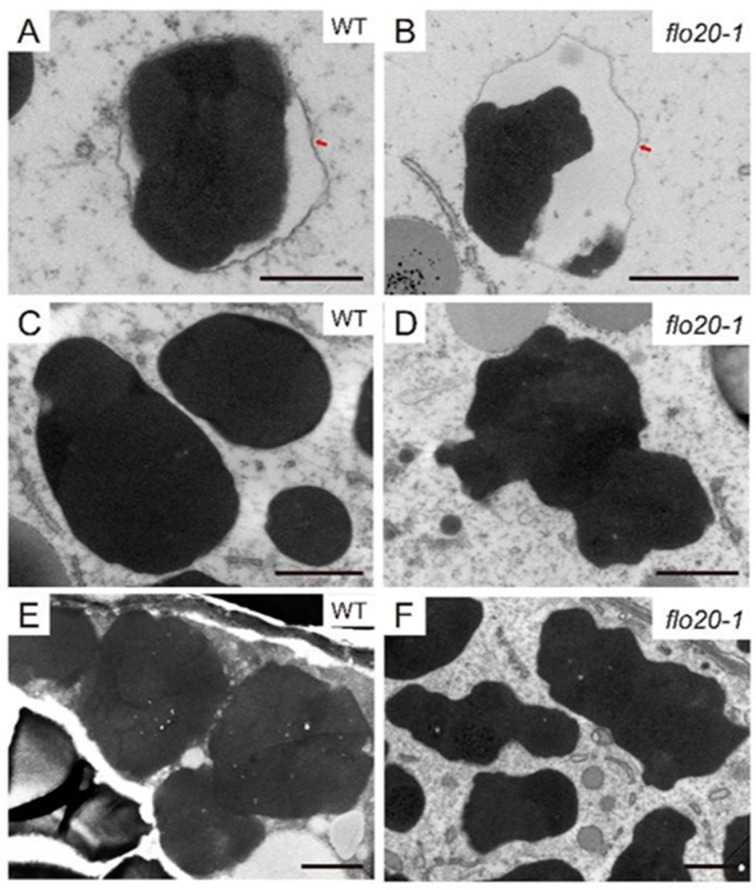
Ultrastructure of PBIIs during the development of the endosperms of WT and *flo20-1* mutants. (**A**–**F**) Transmission electron microscopy of PBIIs in subaleurone cells of developing seeds at 9 DAF (**A**,**B**), 15 DAF (**C**,**D**), and 21 DAF (**E**,**F**). Red arrows indicate the vacuolar membrane. Bars in (**A**,**B**) are 500 nm. Bars in (**C**–**F**) are 1 μm.

**Figure 3 plants-13-00081-f003:**
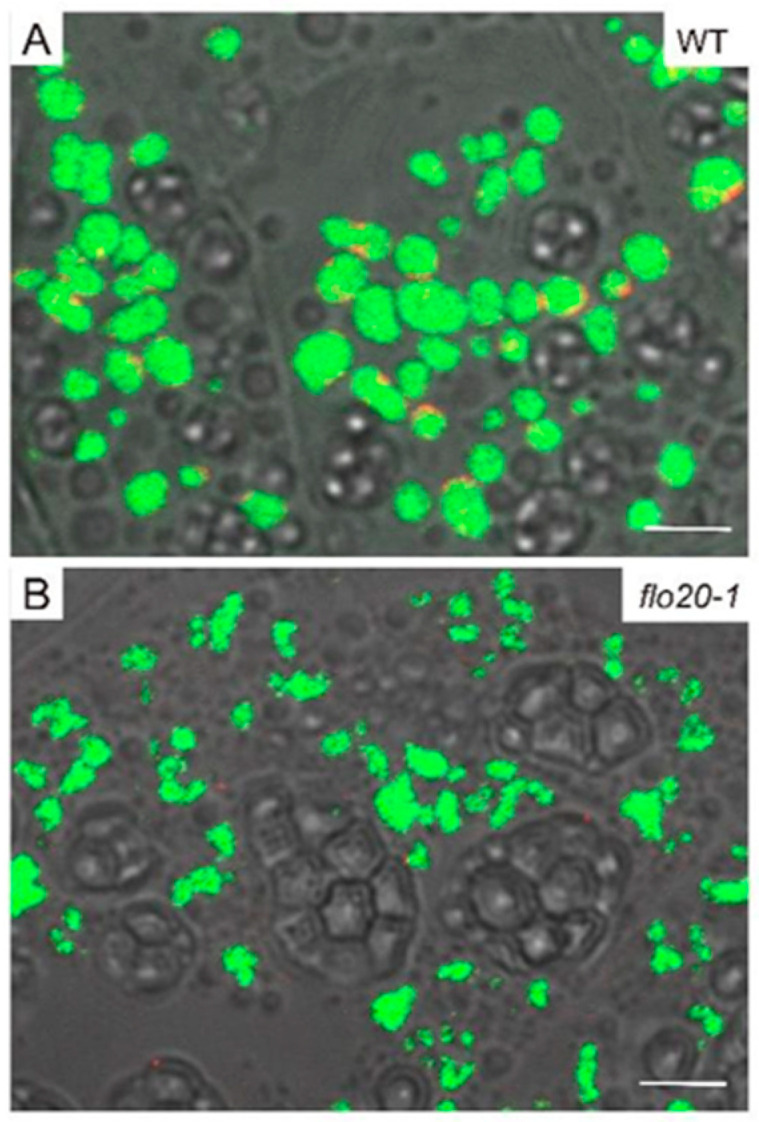
Double immunofluorescence microscopy of glutelin and α-globulin in the developing subaleurone cells of the WT (**A**) and *flo20-1* (**B**) endosperms at 9 DAF. Antigens recognized by the polyclonal anti-glutelin antibodies from rabbit and monoclonal anti-α-globulin antibodies from mice were detected using secondary antibodies labeled with Alexa Fluor 488 (green) and Alexa Fluor 555 (red), respectively. Bars = 5 μm.

**Figure 4 plants-13-00081-f004:**
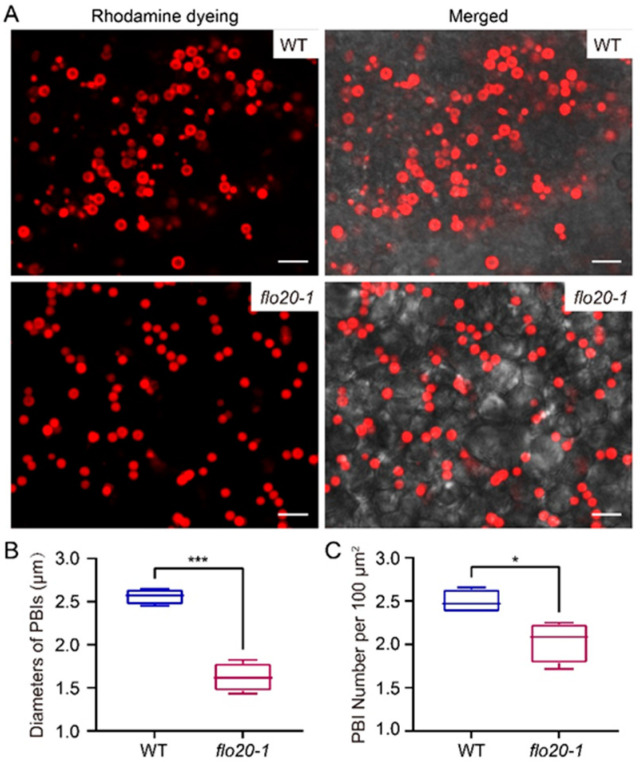
Confocal microscopy observation of PBIs in the subaleurone cells of WT and *flo20-1* mutants. (**A**) Confocal microscopy images of rhodamine-labeled PBIs in WT and *flo20-1* mutant grains. Bars = 5 μm. (**B**,**C**) Measurement of the diameters of PBIs (**B**) and the number of PBIs per 100 μm^2^ (**C**). Values are means ± SD. * *p* < 0.05, *** *p* < 0.001 (*n* = 4 (total of 382 PBIs) for the WT and 4 (total of 279 PBIs) for *flo20-1* in (**B**,**C**); Student’s *t* test).

**Figure 5 plants-13-00081-f005:**
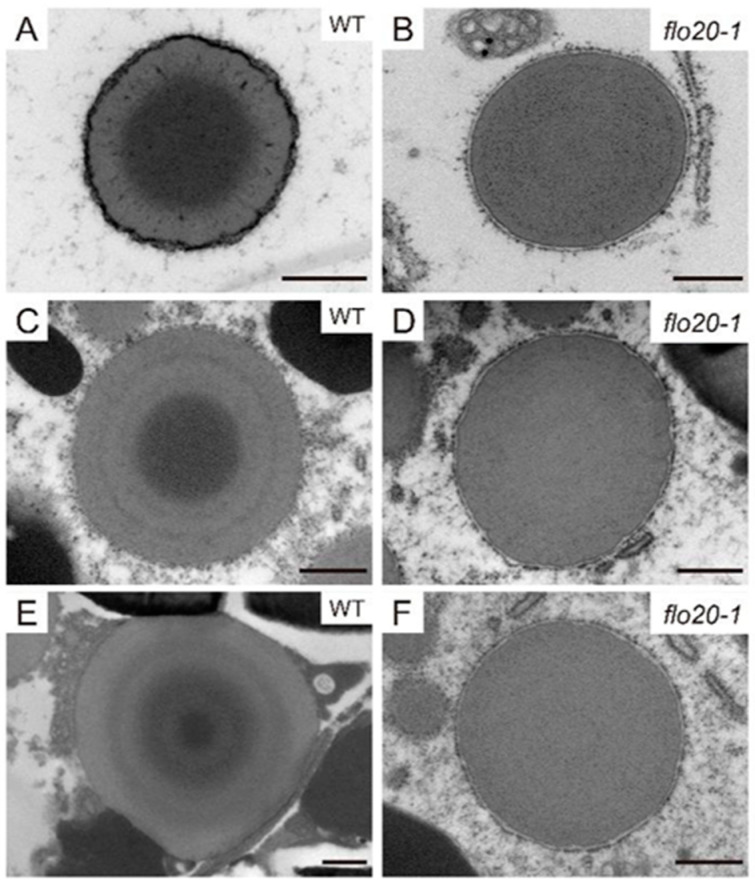
Ultrastructure of PBIs during the development of endosperms of WT and *flo20-1* mutants. (**A**–**F**) Transmission electron microscopy of PBIs in subaleurone cells of developing seeds at 9 DAF (**A**,**B**), 15 DAF (**C**,**D**), and 21 DAF (**E**,**F**). Bars = 500 nm.

**Figure 6 plants-13-00081-f006:**
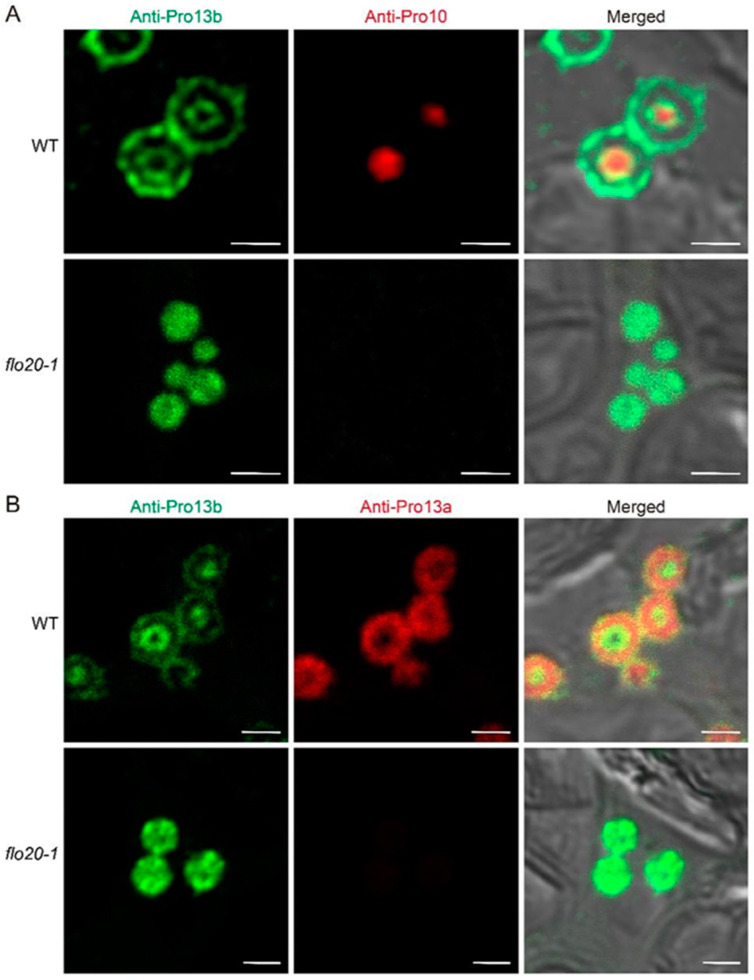
Double immunofluorescence microscopy of prolamins in the subaleurone cells of the WT and *flo20-1* endosperms at 21 DAF. Alexa-Fluor-488-labeled secondary antibodies (green) and Alexa-Fluor-555-labeled secondary antibodies (red) were employed to identify antigens targeted by monoclonal anti-Pro13b antibodies from mouse and polyclonal anti-Pro10 (**A**) or anti-Pro13a (**B**) antibodies from rabbits, respectively. Bars = 2 μm.

**Figure 7 plants-13-00081-f007:**
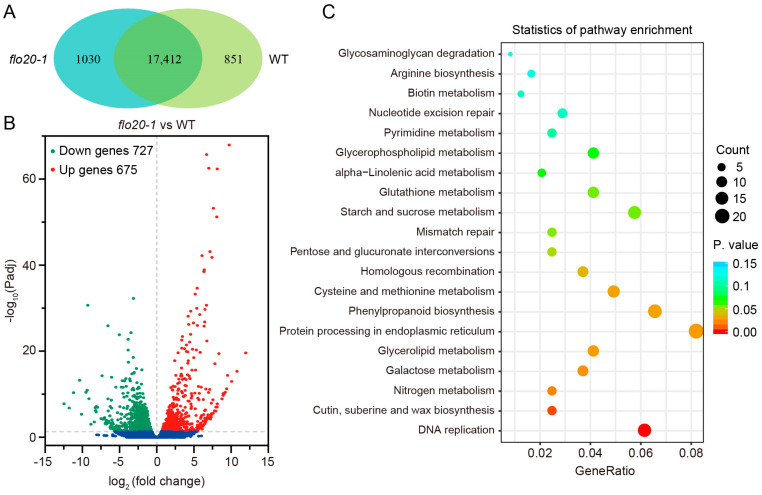
Global transcriptome analysis in 9 DAF endosperm of the WT and *flo20-1*. (**A**) Venn diagram showing the number of genes expressed in the WT, *flo20-1,* or in both genotypes. (**B**) Volcano plot indicating the upregulated and downregulated genes in *flo20-1* relative to the WT. (**C**) KEGG pathway classification for genes showing differential expression in *flo20-1* endosperm compared to the WT.

**Figure 8 plants-13-00081-f008:**
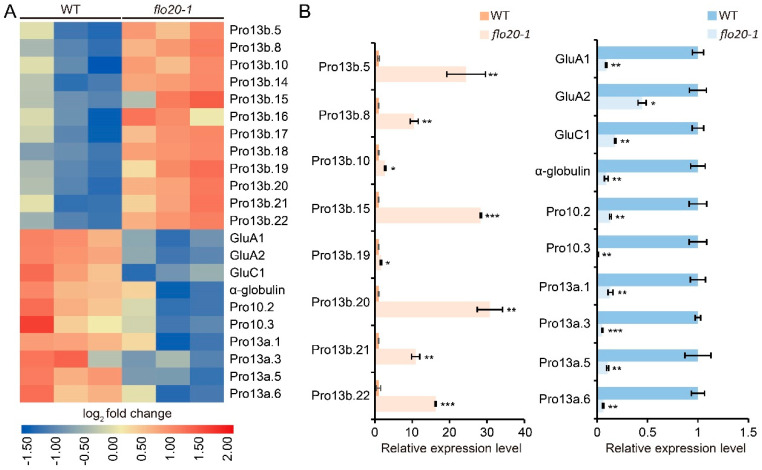
RNA-seq and qRT-PCR analyses to show the expression of genes involved in storage protein synthesis. (**A**) Differentially expressed genes involved in storage protein synthesis in WT and *flo20-1* endosperm at 9 DAF based on RNA-seq data. (**B**) The relative expressions of representative genes were validated through qRT-PCR. For each RNA sample, three biological replicates were performed. Values are mean ± SD. * *p* < 0.05, ** *p* < 0.01, and *** *p* < 0.001 according to Student’s *t* test.

## Data Availability

Data are contained within the article and Appendix A.

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
