# Peer review of "OsSHMT4 Is Required for Synthesis of Rice Storage Protein and Storage Organelle Formation in Endosperm Cells"

_plants, 2023, doi:10.3390/plants13010081_

Round 1

Reviewer 1 Report

Comments and Suggestions for Authors

Review of plants-2769748

In the article, two rice endosperm (mature wild-type WT and floury 20-1 mutant flo20-1) were detail investigated from the point of view of storage proteins important and essential on seed germination and seedling growth during developing WT endosperm at 9, 15 and 21 DAF. The knowledge about storage protein composition (prolamins, glutelin subunits, globulin polypeptides, regulatory factor OsSHMT4, amorphous structure of PBII, abnormal morphological structure of PBI, transcriptome sequencing analysis, SDS-PAGE and immunoblot analyses, and analysis of DEGs related to the SSPs synthesis significantly expanded knowledge about the role the expression levels of several structural genes and their expression on the alteration of transcription levels of storage protein structural genes and the abnormal synthesis and accumulation of storage protein. Expanded understanding about the function of genes in changes of storage proteins was enabled using media analysis techniques, like SSPs extraction, immunoblot analyses, analyses using monoclonal and polyclonal antibodies, rhodamine staining microscopy analyses, double immunofluorescence analysis, and RNA extraction and RNA-seq analysis or quantitative real-time PCR (qRT-PCR) validation of DEGs.

To the article, I have next comments and recommendations:

·         L. 386: better β-sulfanylethanol or 2-sulfanylethanol according to IUPAC nomenclature.

·         All applied equipment and reagents should be carefully completed with the name of the producer, town, and country (state) – check and complete the missing date especially in section Material and Methods: L. 398-400: TWEEN20 (producer, town, country), ECL Plus reagent (Odyssey-Fc LI-COR, town, country), ChemiDoc System (BIO-RAD, town, country); L. 407-408: Anti-EF-1α (Agrisera, AS10934, town, country…); L. 414-415: oscillating microtome (LEICA VT1200S, town, country); L. 417-418: rhodamine (Sigma 83689, town, country); L. 420: LSM980 laser scanning confocal microscope (Carl Zeiss, town, country), L. 428-429: Alexa fluor 488 (town, country), Alexa fluor 555 (town, country); (Invitrogen, town, country); L. 438: Hitachi H7700 transmission electron microscope (town, country); L. 441: Hitachi H7700 transmission electron microscope (town, country); L. 452-453: PrimeScript Reverse Transcriptase Kit (TaKaRa, town, country), the SYBR Premix Ex Taq (should be Takara or TaKaRa?, town, country).

Author Response

We acknowledge the reviewer’s helpful comments and recommendations, which are valuable for improving our manuscript.

  1. 386: better β-sulfanylethanol or 2-sulfanylethanol according to IUPAC nomenclature.

Reply: Thank you for your comments. Following your suggestion, β-sulfanylethanol has been changed into 2-sulfanylethanol (Line 395). According to the IUPAC nomenclature, the preferred name for the compound with a thiol (sulfhydryl) group at the 2-position in ethanol would be "2-sulfanylethanol." The "2-" indicates the position of the sulfur atom in the molecule. Therefore, "2-sulfanylethanol" is the correct IUPAC name for this compound.

  1. All applied equipment and reagents should be carefully completed with the name of the producer, town, and country (state) – check and complete the missing date especially in section Material and Methods: L. 398-400: TWEEN20 (producer, town, country), ECL Plus reagent (Odyssey-Fc LI-COR, town, country), ChemiDoc System (BIO-RAD, town, country); L. 407-408: Anti-EF-1α (Agrisera, AS10934, town, country…); L. 414-415: oscillating microtome (LEICA VT1200S, town, country); L. 417-418: rhodamine (Sigma 83689, town, country); L. 420: LSM980 laser scanning confocal microscope (Carl Zeiss, town, country), L. 428-429: Alexa fluor 488 (town, country), Alexa fluor 555 (town, country); (Invitrogen, town, country); L. 438: Hitachi H7700 transmission electron microscope (town, country); L. 441: Hitachi H7700 transmission electron microscope (town, country); L. 452-453: PrimeScript Reverse Transcriptase Kit (TaKaRa, town, country), the SYBR Premix Ex Taq (should be Takara or TaKaRa?, town, country).

Reply: Thank you for your valuable comment. Following your suggestion, specific information about equipment and reagents have been completed in our revised manuscript.

The corrected statements are below:

Line 407-408: TWEEN20 (VWR Chemicals, Radnor, America)

Line 409: ECL Plus reagent (Odyssey-Fc LI-COR, Lincoln, America)

Line 410: ChemiDoc System (BIO-RAD, Hercules, America)

Line 418: Anti-EF-1α (Agrisera, AS10934, Uppsala, Sweden)

Line 425-426: oscillating microtome (LEICA VT1200S, Wetzlar, Germany)

Line 429: rhodamine (Sigma-Aldrich 83689, St. Louis, America)

Line 431-432: LSM980 laser scanning confocal microscope (Carl Zeiss, Oberkochen, Germany)

Line 441-442: Alexa fluor 488 (green) and/or Alexa fluor 555 (red) (1:200 dilution) conjugated secondary antibodies (Invitrogen, Carlsbad, America)

Line 452-453: Hitachi H7700 transmission electron microscope (Tokyo, Japan)

Line 456: RNA Prep Pure kit (TIANGEN Biotech, Beijing, China)

Line 467: PrimeScript Reverse Transcriptase Kit (TaKaRa, Kyoto, Japan)

Line 468: ABI7500 Real-time PCR system (Thermo Fisher Scientific, Waltham, America)

Line 469: SYBR Premix Ex Taq (TaKaRa, Kyoto, Japan)

Reviewer 2 Report

Comments and Suggestions for Authors

Dear Authors,

Manuscript title:

OsSHMT4 is required for the synthesis of rice storage protein and two storage organelles formation in endosperm cells

This manuscript addresses the effect of Os SHMT 4 mutation in rice on the change of storage protein in the rice endosperm.  This manuscript is very well written, and the results are adequately presented. Interesting for the broad scientific community.

The discussion is well-organized and addresses all the author's findings.

There are small details that can be corrected before the manuscript is published.

Abstract:  Line 21, please write a full name for the PBII abbreviation here.

Keywords: please do not repeat the same word from the title.

Introduction:  well written. Please clearly formulate the goals of this study.

Lines 85-88 are better suited to the section Conclusions.

Results: Well structured and presented.

Discussion: Well-written and connected with the author’s findings.

M&M:

Line 380: Please add the information about the geographic location where the rice donor plants were growing.

This exciting manuscript describes an interesting topic relevant to many scientists working not only with rice.

15.12.2023

Author Response

We acknowledge the reviewer’s constructive comments and recommendations, which are helpful for improving our manuscript.

  1. Abstract: Line 21, please write a full name for the PBII abbreviation here.

Reply: Thank you for your careful comments. As you suggested, we have supplemented the abbreviation in our revised manuscript (Line 21-22).

  1. Keywords: please do not repeat the same word from the title.

Reply: Thank you for your valuable comment. Following your suggestion, the keywords in the manuscript have been revised to include: seed storage protein; flo20-1; PBII; PBI; transcription levels.

  1. Introduction: well written. Please clearly formulate the goals of this study.Lines 85-88 are better suited to the section Conclusions.

Reply: Thank you for your comments. We have revised this description into “we demonstrated the morphological structures of PBIs and PBIIs in flo20-1 endosperm by cytological experiments to explicit the effects of OsSHMT4 on seed storage proteins and performed RNA-seq analysis to elucidate underlying causes” in our new manuscript (Line 89-91).

  1. Line 380: Please add the information about the geographic location where the rice donor plants were growing.

Reply: Thank you very much for your careful comments. We have now added the information about the geographic location where the rice donor plants were growing in the revised manuscript (Line 390-391).

The description is below:

“Developing seeds and mature seeds were taken from the plants growing in paddy fields during the normal growing season in Beijing.”
